# Declines over the last two decades of five intertidal invertebrate species in the western North Atlantic

Peter S. Petraitis [1✉] & S. R. Dudgeon [2✉]

Climate change has already altered the environmental conditions of the world's oceans. Here we report declines in gastropod abundances and recruitment of mussels (*Mytilus edulis*) and barnacles (*Semibalanus balanoides*) over the last two decades that are correlated with changes in temperature and ocean conditions. Mussel recruitment is declining by 15.7% per year, barnacle recruitment by 5.0% per year, and abundances of three common gastropods are declining by an average of 3.1% per year (*Testudinalia testudinalis*, *Littorina littorea*, and *Nucella lapillus*). The declines in mussels and the common periwinkle (*L. littorea*) are correlated with warming sea temperatures and the declines in *T. testudinalis* and *N. lapillus* are correlated with aragonite saturation state, which affects rates of shell calcification. These species are common on shores throughout the North Atlantic and their loss is likely to lead to simplification of an important food web on rocky shores.

[1] Department of Biology, University of Pennsylvania, Philadelphia, PA 19104-6018, USA. [2] Department of Biology, California State University, Northridge, CA 91330-8303, USA. ✉email: ppetrait@sas.upenn.edu; steve.dudgeon@csun.edu

Predictions about how climate change will affect marine ecosystems have been largely based on extrapolations from short-term experiments of physiological changes or retrospective surveys[1–5]. While the effects of climate change on the oceans are predicted to lead to declines in consumer biomass at mid and higher trophic levels and simplification of food webs[6–8], there are few long-term, longitudinal surveys of populations in the wild to support these predictions[9,10]. Here, we provide evidence for decadal declines of five common benthic species (the tortoiseshell limpet, *Testudinalia testudinalis*, the common periwinkle, *Littorina littorea*, the dogwhelk, *Nucella lapillus*, the blue mussel, *Mytilus edulis*, and the barnacle, *Semibalanus balanoides*) on rocky shores of the NW Atlantic Ocean, and link these declines to changes in oceanic conditions in the Gulf of Maine due to climate change. Because of climate change, the Gulf of Maine has recently warmed faster than 99.9% of the global oceans[11] and has a relatively low pH and aragonite saturation state ($\Omega_{AR}$), rendering the Gulf of Maine poorly buffered against ocean acidification[12]. Changes in water temperature, pH, and $\Omega_{AR}$ affect the development, growth, calcification, and survivorship of marine organisms[1,3], and in particular, changes in ocean pH and $\Omega_{AR}$ can slow the calcification of molluscan shells[4]. Our observations are one of the few continuous long-term studies to document declines in situ and to link those declines to climate change. The declines are surprising in that the five species have been intensively studied for more than 50 years.

These species are key members of intertidal ecosystems throughout the North Atlantic, and their declines will have profound effects on intertidal food webs. Mussels and barnacles are dominant competitors for space on hard surfaces and major prey items for many invertebrate and vertebrate predators. Dogwhelks, periwinkles, and limpets are not only important consumers but also important prey. These species were the basis of one of the first diagrams of a food web[13,14], and have been used as model organisms in ground-breaking ecological experiments on competition, predation, and community structure[15–18]. We hypothesize the declines will cause structural shifts in the food web by enhancing primary producers (i.e., seaweeds and microalgae) and depressing the importance of invertebrate consumers. Continued losses of mid-trophic level consumers will likely lead to simplification of this iconic food web with profound implications for secondary production and stability of rocky shore ecosystems throughout the North Atlantic[6,19,20].

## Results

Our observations come from a larger long-term experiment for which we have been sampling the abundances of gastropods and recruitment by barnacles and mussels since 1997. The data are from 12 control plots that are distributed across four bays on Swans Island, Maine, USA. The control plots have not been manipulated, are in their natural state, and any pair are >0.5 km apart. From 1997 to 2018, recruitment of mussels (*M. edulis*) and barnacles (*S. balanoides*), and abundances of periwinkles (*L. littorea*), limpets (*T. testudinalis*), and dogwhelks (*N. lapillus*) declined (Fig. 1 and Supplementary Tables 1 and 2). Bayesian estimates of per year rates of decline, ranked from fastest to slowest, are: 15.7% for mussel recruitment (95% credible interval: 13.3–18.2%), 6.0% for limpet abundance (1.9–9.5%), 5.0% for barnacle recruitment (0.1–9.7%), 3.8% for dogwhelk abundance (2.0–6.5%), and 3.2% for periwinkle abundance (2.2–4.3%). Changes in abundance of smooth periwinkle (*L. obtusata*) are indistinguishable from zero (an average annual increase of 1.0%, CI: −0.9 to +2.7%).

Declines in mussels and the common periwinkle, both of which have planktonic larvae, are correlated with warmer oceanic temperatures in the Gulf of Maine (Supplementary Tables 3–5 and Fig. 2). The decline in mussel recruitment is best explained by water temperature in August, which is when mussel larvae are present in the water column. August water temperature has increased at a rate of 0.085 °C year$^{-1}$ (95% credible interval: 0.022–0.148; Supplementary Table 6 and Supplementary Fig. 1). For every half-degree increase in temperature, which occurs approximately every 6 years, there is approximately a tenfold decline in mussel recruitment (Fig. 2b; $P < 0.0002$). The decline in the common periwinkle (*L. littorea*) is best explained by a 1-year lag in the annual average water temperature (Fig. 2a; $P < 0.0001$). Note that we usually sampled abundances in July of each year, and for annual average water temperature, we used the average of monthly mean water temperatures from June of the preceding year to July of the current year. We saw a loss of 50 individuals per m$^2$ for each 1 °C increase. The 1-year lag suggests vulnerability in the planktonic stage takes at least 1 year to be evident as a decline in adult populations. The annual mean water temperature increased at a rate of 0.069 °C year$^{-1}$ (95% credible interval: 0.005–0.128; Supplementary Fig. 1 and Supplementary Table 6), and a one-degree increase would take 14.5 years. In contrast, a decline in the barnacle *S. balanoides*, which also has a long planktonic phase, could not be linked to changes in water temperature, pH or $\Omega_{AR}$ (Fig. 2c and Supplementary Table 5).

A 1-year lag in annual $\Omega_{AR}$ is the best predictor of the changes in the remaining gastropod species (Fig. 2d–f). Abundances of the limpet *T. testudinalis*, the dogwhelk *N. lapillus* and the smooth periwinkle *L. obtusata* decline with larger values of $\Omega_{AR}$. Gastropods show lower rates of shell calcification at lower $\Omega_{AR}$ values[2–4], yet we observed larger abundances occurring at smaller values for $\Omega_{AR}$. Notably, $\Omega_{AR}$ in the Gulf of Maine covaries with not only acidification, warming, and salinity but also local nearshore conditions[21], and so we hypothesize $\Omega_{AR}$ may be a proxy for the combined and adverse effects of acidification, warming, and salinity.

Two additional observations suggest local, small-scale characteristics, such as the aspect of the shore and tidal flow within specific bays, also play a role. First is a large amount of spatial variability. Between 20 and 50% of the total variance in abundances and recruitment is due to variation among sites (Supplementary Table 7). Second, the residuals from the predicted declines are correlated (Supplementary Table 8). Four of the six pairwise correlations of abundances among the gastropods are significant and range from +0.154 to +0.418 ($P < 0.01$ in all cases). There are also clear peaks in 2006 in abundances of *N. lapillus*, *L. littorea*, and *L. obtusata*, and in barnacle recruitment (Fig. 2a, c, d, f, respectively). These are more likely due to mesoscale processes rather than localized conditions, but we have no evidence for a potential driver.

## Discussion

The rates of declines we have observed are sobering. Over the last two decades, abundances of limpets, periwinkles, and dogwhelks have declined by at least 50%, and we predict over the next 10–20 years, there will be an additional 50% reduction in (Fig. 1 and Supplementary Table 2). More alarming is the loss of mussel beds, which we think has been driven by nearly a 16% decline in recruitment per year.

Southward's data on the abundance of *S. balanoides* and other barnacle species from 1951 to 1990 are the only other comparable data from the north Atlantic Ocean[22]. While this remarkable set of data has been expanded by other researchers and used in a number of studies, none report a decline in barnacle recruitment or abundances[5,23–25]. We suspect the one reason why we have seen such dramatic declines not seen by Southward and his

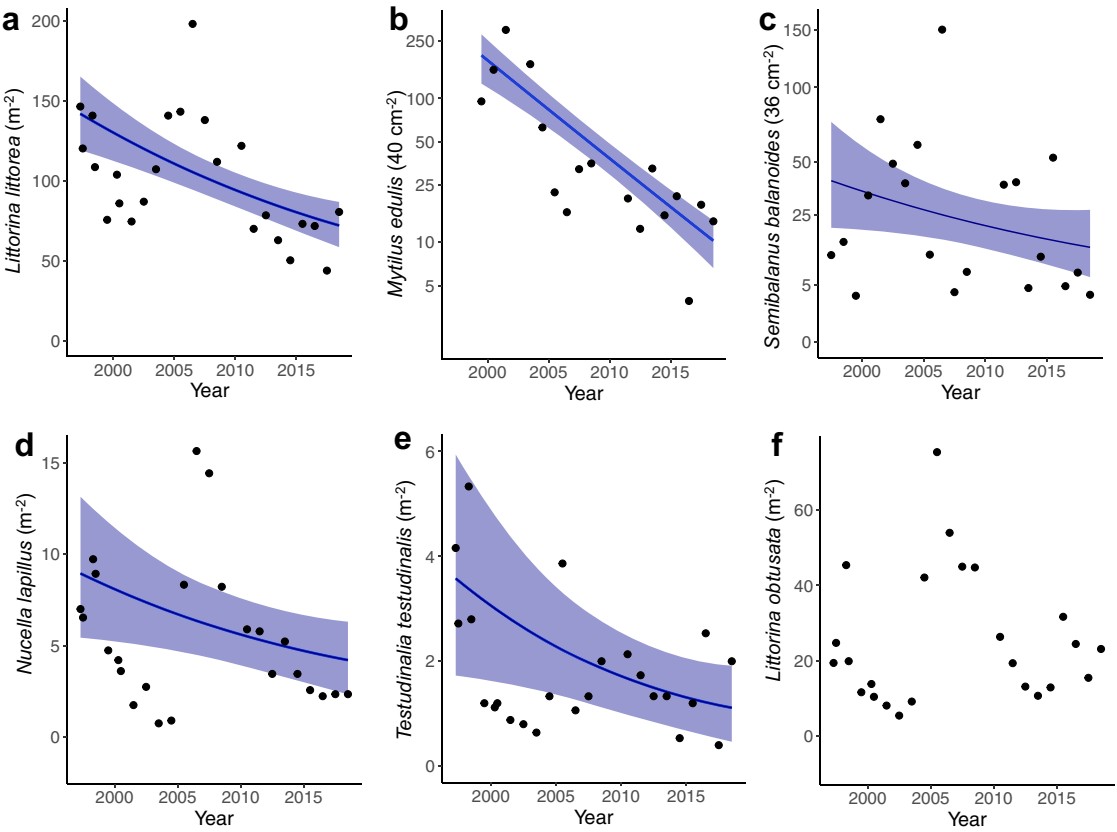

**Fig. 1 Changes in snail abundances and recruitment of mussels and barnacles versus year.** Curves are Bayesian estimates of Poisson regressions, and shaded regions are 95% credible intervals (Supplementary Table 1). **a** Mean abundance of the common periwinkle (*L. littorea*). **b** Mean recruitment of mussels (*M. edulis*); data plotted on a $\log_{10}$ scale. **c** Mean recruitment of barnacles (*S. balanoides*); data plotted on a square-root scale. **d** Mean abundance of the common dogwhelk (*N. lapillus*). **e** Mean abundance of the common tortoiseshell limpet (*T. testudinalis*). **f** Mean abundance of the smooth periwinkle (*L. obtusata*). The estimate of slope for *L. obtusata* is indistinguishable from zero, and so data are plotted without a curve or shaded region.

co-workers is the environmental shifts in the Gulf of Maine due to climate change have been more rapid and extreme than elsewhere[11,12,21].

Our results suggest the effects of climate change on molluscs depend on larval life history. The species with pelagic larvae (mussels and the common periwinkle) are affected by temperature, while species with larvae that are not planktonic (dogwhelks and the smooth periwinkle), or that have a very short planktonic stage (the tortoiseshell limpet), are affected by a combination of factors for which $\Omega_{AR}$ may serve as a proxy (e.g., acidification, warming, increased salinity). For molluscs with pelagic larvae, warmer oceanic waters affect metabolic rates, development times, and the composition of the plankton, on which larvae depend on for food[3]. For dogwhelks, smooth periwinkles, and tortoiseshell limpets, it is quite likely reproductive success and abundance depend on conditions in the local intertidal habitat, as it is well known that changes in acidification, temperature, and salinity can be quite extreme in nearshore environments[4].

In contrast, the decline in barnacle recruitment could not be linked to oceanic conditions, and the intercept-only model was the best-supported model (Supplementary Table 5). Lab studies have shown a combination of warming and lower pH can slow the embryonic development of *S. balanoides*[26], but we were unable to detect a relationship with either pH or sea temperature in situ. It is notable that barnacles, which are crustaceans and unlike molluscs, do not begin calcification as pelagic larvae. The decline in barnacle recruitment, therefore, may be less affected by changes in pH and $\Omega_{AR}$. In addition, the lack of a relationship between recruitment and ocean temperature may be due to the

timing of when adult barnacles release larvae into the plankton. In the Gulf of Maine, larvae are usually released in early spring and are linked to the timing of the spring phytoplankton bloom[27]. Moreover, our estimate of the rate of change in ocean temperature in March shows a warming trend, although the credible interval is quite large and encompasses estimates of cooling trends (mean rate of 0.062 °C year$^{-1}$; CI: −0.048–0.179 °C year$^{-1}$; see Supplementary Table 6).

The relationships between warming ocean temperatures and declines in species with pelagic larvae should not be surprising given how quickly temperatures are rising in the Gulf of Maine. A modeling effort using sea surface temperature (SST) data throughout the Gulf of Maine shows an increase of 0.03 °C year$^{-1}$ between 1982 and 2003 and then a dramatic jump to a rate of 0.23 °C year$^{-1}$ starting in 2004[11]. These rates are among the most rapid increases seen worldwide. Even so, our estimates of the rate of increase in ocean temperature are even larger (Supplementary Table 6). Our estimate, based on data from an offshore buoy close to our sites, is 0.069 °C year$^{-1}$ (CI: 0.005–0.128 °C year$^{-1}$) from 2002 to 2018. However, we do not see a jump in the rate when we use only the data from 2004 to 2014. Our estimate for this decade is 0.126 °C year$^{-1}$, and the credible interval includes not only the published estimate of 0.23 °C year$^{-1}$ but also cooling of ocean temperatures (CI: −0.009–0.249 °C year$^{-1}$).

The Gulf of Maine has also seen declines in pH and aragonite saturation state ($\Omega_{AR}$) in the Gulf of Maine; from 1981 to 2014, and the rates of decline for pH and $\Omega_{AR}$ have been 0.0018 year$^{-1}$ and 0.049 year$^{-1}$, respectively[12,21]. However, our observations of the links between the changes in water chemistry and declines in

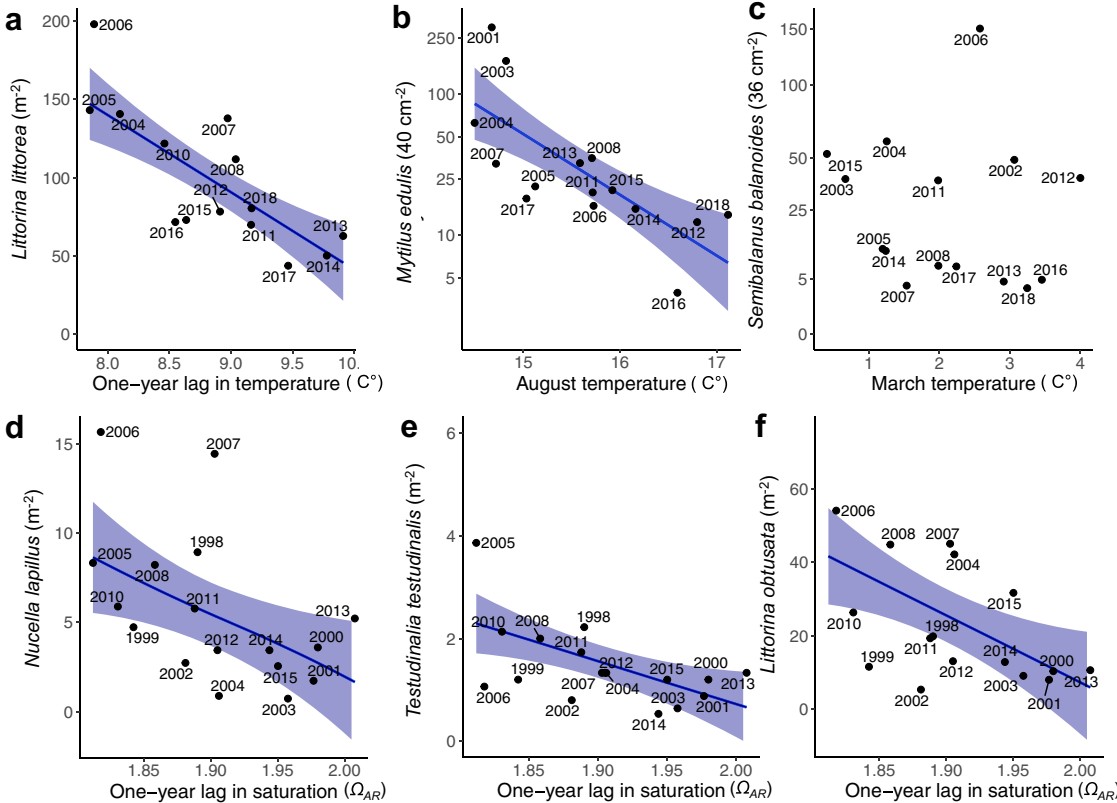

**Fig. 2 Changes in snail abundances and recruitment of mussels and barnacles versus environmental conditions.** Curves are based on pooled estimates from ten iterations of the imputation of missing values, and shaded regions are the 95% confidence limits (Supplementary Table 3). **a** Mean abundance of the common periwinkle (*L. littorea*) versus a 1-year lag in mean annual ocean temperatures. **b** Mean recruitment of mussels (*M. edulis*) versus mean August ocean temperatures; data plotted on a $\log_{10}$ scale. **c** Mean recruitment of barnacles (*S. balanoides*) versus mean March ocean temperatures; data plotted on a square-root scale. The intercept-only model is best supported (Supplementary Table 5), and so data are plotted without a curve or shaded region. **d** Mean abundance of the common dogwhelk (*N. lapillus*) versus a 1-year lag in mean annual saturation ratios. **e** Mean abundance of the common tortoiseshell limpet (*T. testudinalis*) versus a 1-year lag in mean annual saturation ratios. **f** Mean abundance of the smooth periwinkle (*L. obtusata*) versus a 1-year lag in mean annual saturation ratios.

species abundances should be viewed with caution for several reasons. The values for pH and $\Omega_{AR}$, which were extracted from the literature, are average yearly estimates across the entire Gulf of Maine. In the Gulf of Maine, values for pH and, in particular, $\Omega_{AR}$, show large annual and decadal variation, and covary with changes in salinity, total alkalinity, and water temperature[21]. Moreover, freshwater inflows from rivers also affect pH, salinity, and total alkalinity. Even so, our findings raise interesting questions and suggest declines in species abundances, and recruitment may vary from site to site because of very localized differences in water chemistry.

The declines in mussel and barnacle recruitment are likely to have major impacts because both are important prey and foundation species of rocky intertidal ecosystems throughout the western North Atlantic Ocean[15,17]. Both species can occupy broad swathes of the shore, and their losses as common prey items will affect consumers. Mussels have already disappeared from many intertidal shores throughout the North Atlantic[20,28]; a loss that has been widely attributed to the recent rise in the predatory green crab, *Carcinus maenas*[29,30]. Yet, our results suggest recruitment failure associated with processes in the water column is also playing an important role in the decline of mussels throughout the North Atlantic.

The loss of gastropods, which are important consumers, will affect a number of trophic interactions and likely lead to a simplification of the food web[19]. The dogwhelk preys upon barnacles

and mussels, and so the loss of barnacles and mussels may hasten the decline of dogwhelks. Herbivorous gastropods (periwinkles and limpets) strongly affect the distribution of macroalgae, especially ephemeral species[16,17,31]. Unlike what has been predicted for the shores of Great Britain and western Europe, it is unlikely that southern species will expand their ranges northward to fill the gaps[10,32]. Recolonization of rocky intertidal shores in the Gulf of Maine after the last glacial retreat was by European species rather than range expansions of southern species[33,34]. Moreover, most of the shoreline south of Cape Cod, MA, does not include rocky shores and so there is simply not the species pool to support an expansion from the south. One interesting possibility is that the loss of the common periwinkle *L. littorea* may return the food web to its state prior to 1860. The common periwinkle, which is now the most common gastropod herbivore from Nova Scotia to Cape Cod, MA, was not found in the Gulf of Maine prior to 1860[35].

There may very well be other unanticipated impacts. For example, there may be profound effects not only on the flow of carbon through food webs on rocky intertidal shores throughout the North Atlantic but also the export of carbon to offshore ecosystems. The loss of filter feeders (mussels and barnacles) will certainly break the benthic–pelagic coupling between phytoplankton and fecal input into the benthos[36,37]. Moreover, while increases in dissolved $p$CO$_2$ and water temperatures due to climate change will fuel increases in primary production[38], the loss

of consumers may reduce the export of carbon as the links among phytoplankton, filter feeders, and consumer are broken. This is only one of many potential effects, and we anticipate there will be many more surprises in how nearshore ecosystems will respond to the combined effects of climate change and the loss of species.

## Methods

**Description of study plots, data collection, and data availability**. Data were collected from 12 unmanipulated control plots on Swans Island (44°10' N, 68°25' W), a small island (approx. 36 km² in area) in the Gulf of Maine, USA. The control plots are part of a larger, long-term study started in 1996, and each control plot is located at one of the 12 sites on Swans Island. Descriptions of the study sites, the location of each plot, methods of data collection and experimental design, and data for recruitment from 1997 to 2012 and for abundances from 1996 to 2008 are available elsewhere[18,39–41]. The 12 sites are spread over four different bays on Swans Island, and most are more than 1 km apart. The two closest sites are slightly <1 km apart. All plots are in the mid intertidal zone, which is dominated by stands of the fucoid rockweed, *Ascophyllum nodosum* interspersed with small patches of the rockweed *Fucus vesiculosus*. The species composition of invertebrates and algae at these sites is typical of sheltered rocky shores in the north Atlantic[42].

Recruitment data for *Semibalanus balanoides* and *Mytilus edulis* have been collected annually since 1997. Abundances of the four most common gastropods (*Testudinalia testudinalis*, *Littorina littorea*, *Littorina obtusata*, and *Nucella lapillus*) have been sampled at least once a year since 1996. Note that until 2010, *Testudinalia testudinalis* was previously named *Tectura testudinalis*[43].

Environmental data for pH, aragonite saturation ratio ($\Omega_{AR}$), and temperature were extracted from published papers and online databases. Yearly averages in pH and $\Omega_{AR}$ from 1997 to 2014 were taken from published figures[21]. Monthly average temperatures at 1 m depth and from 2002 to 2018 were downloaded from NERACOOS for the three buoys that are the closest to Swans Island (http://neracoos.org/datatools/climatologies_display). The buoys are E01 (Central Maine Shelf), F01 (Penobscot Bay), and I01 (Eastern Maine Shelf). Averages among the buoys were highly correlated, and all three pairwise correlations were greater than 0.942. Buoy F01 was missing the fewest observations (April 2007, May 2007, and October 2008), and so these data were used for analysis. Monthly and yearly averages for recruitment, abundances, and environmental data are given in Supplementary Data 1; raw data and R script for analyses are available online[44].

**Regressions of changes in recruitment and abundance over time**. Regressions were done assuming Poisson error distribution using the R package MCMCglmm[45]. The package fits generalized linear mixed models using Markov chain Monte Carlo techniques. The date of sampling is a fixed factor, and the 12 sites are treated as a random effect. We set the thinning rate to 100, the burn-in number to 6000, and the total number of iterations to 206,000. This gives an effective sampling of ~2000 for all parameters in most regressions. We used default priors. Sampling was done annually, except in 1997, 1998, and 2000 when sites were sampled in spring and summer. Spring samples were coded as occurring in April (e.g., 1997.3), and summer samples were coded as occurring mid-year (e.g., 1997.5). Data for all species are converted to standardized units and treated as integers (number per m² for gastropods, per 36 cm² for barnacles, and per 40 cm² for mussels). Units are based on the area or surface used to sample each species[40,41]; for example, the tile for sampling barnacle recruitment is circular with an area of 39.6 cm².

In three cases, data were transformed or dropped because models using the full dataset failed to converge. Data for barnacle (*S. balanoides*) recruitment were square-root transformed. For mussel recruitment (*M. edulis*), 1997 data were dropped because of extremely large values (average 1161 per 40 cm², range: 32–5424). For limpet abundance (*T. testudinalis*), data from two sites were dropped because limpets were absent or very rare from 2000 to 2018, and the datum for one site in 1998 for which abundance was >10 per m².

**Regressions of changes in water temperature over time**. Data for the regressions were the monthly average temperatures from Buoy F01 at 1 m depth and from 2002 to 2018. Regressions were done using yearly maximum, minimum, and median water temperatures, which usually occurred in August, February, and May or November, respectively. Regressions were done assuming Gaussian error distribution using the R package MCMCglmm[45]. Month and year were entered as a continuous number (e.g., February 2002 was coded as 2002.125). The thinning rate was set to 100, the burn-in number to 6000, and the total number of iterations to 206,000. This gives an effective sampling of 2000. We used default priors.

**Regressions of changes in recruitment and abundance versus environmental conditions**. The relationships between environmental variables and the abundances of snails and the recruitment of mussels and barnacles were assessed in two steps. First, corrected Akaike information criteria (AICc), deltas, and Akaike weights were used to assess the best model containing one or two environmental variables[46]. Entries in Supplementary Data 1 were used as the data, and years (i.e., rows) with missing values were dropped. The dredge function in MuMIn package

was used to analyze all possible models[47]. The best-supported models are shown in Supplementary Tables 4 and 5.

The second step involved imputing the missing values and using the imputed values to compile pooled estimates of the best-supported model. Pooled estimates were based on ten iterations of imputation. Missing values were imputed using the R package Amelia[48], and the pooled estimates were calculated using the R package Zelig[49,50]. Pooled estimates are given in Supplementary Table 3 and were used to make Fig. 2. Note that estimates in Supplementary Table 3 differ from estimates in Supplementary Tables 4 and 5.

Because sampling of abundances and recruitment were done at different times of the year, different subsets of the environmental data were used for the dredge function. Snail abundances were usually sampled between early July and early August, and thus for water temperature, we used the yearly average based on the previous year (e.g., for densities sampled in July 2001,) we used the yearly average based on data from July 2000 to June 2001. We used the current year averages for pH and saturation ratio. We also expected changes in densities might lag behind changes in the environment, and so included lags of 1 and 2 years. For the recruitment data, we used monthly averages from the current year for temperature and the current year data for pH and saturation ratios. Recruitment surfaces for barnacles were usually put in the water in late March-early April and picked up in late May; the dredge model included monthly temperature averages for February, March, April, and May. Recruitment pads for mussels were placed out in late May and picked up in late August-early September; the dredge model included monthly temperature data for May, June, July, and August.

**Statistics and reproducibility**. A sampling of abundances and recruitment was done 24 times between 1997 and 2018 at 12 sites for a possible sample size of 288 observations per species. Sample sizes for abundances of all gastropods except limpets were 288 observations per species. There were 49 outliers in the limpet data (*N* = 239 observations); at two sites were individuals were extremely rare (usually zero counts), and for one site in 1998, the mean count was >10. There are missing data for the recruitment of barnacles (*N* = 245) and mussels (*N* = 211). Monthly means for ocean temperature data from August 2001 to December 2018 are from NERACOOS buoy F01; there are three missing means (*N* = 206 out of a possible total of 209). Yearly means for pH and aragonite saturation ratio ($\Omega_{AR}$) from 1997 to 2014 are from the literature[21] (*N* = 18 for both variables). Poisson regressions of abundances and recruitment versus time were done Markov chain Monte Carlo techniques for generalized linear mixed models with time as a continuous variable and sites as a random factor[45]. The best environmental predictors (temperature, pH, and $\Omega_{AR}$) of changes in abundances and recruitment using multimodel inference[46,47], and missing values were imputed[48,49]. See "Methods" section for more details about the data and statistical analyses.

**Reporting summary**. Further information on research design is available in the Nature Research Reporting Summary linked to this article.

## Data availability

The data sets generated and analyzed during the current study are archived and available online via the portal for the Environmental Data Initiative[44]. Three csv data files are available, which are: abundance and recruitment data; monthly means for ocean temperatures from August 2001 to December 2018, and yearly means for pH and aragonite saturation ratio ($\Omega_{AR}$) from 1997 to 2014.

## Code availability

R scripts used to analyze data are archived and available online via the portal for the Environmental Data Initiative[44]. R scripts were run using RStudio and R version 3.6.1 (2019-07-05). R packages used include MCMCglmm, MuMIn, Amelia, and Zelig.

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

## Acknowledgements

We thank Ms. Erika Rhile for her assistance with the collection and entry of the data, and Professors Maria Byrne and Roberto Salguero-Gómez for reading the paper and providing comments. This material is based upon work supported by the National Science Foundation under Grant Nos. OCE-9529564, DEB-0314980, DEB-1020480, and DEB-1555641.

## Author contributions

P.S.P. and S.R.D. developed the hypothesis and collected the data. P.S.P. conducted the statistical analyses. P.S.P. prepared the initial draft, and both authors edited the paper.

## Competing interests

The authors declare no competing interests.
