## [Peer Review File · Communications Biology]

Reviewers' comments:

Reviewer #1 (Remarks to the Author):

Review

This paper details the decline in recruitment and abundance of 5 key rocky shore species as a result of climate change, specifically temperature and ocean acidification. The analysis is novel, although some of the data has been published in other papers (both part of the biological and the environmental data). It is an interesting study and nice to see biological observations and monitoring being considered in the context of climate change and alongside the more routine observations of the physico-chemical environment. The authors propose that the species with longer planktonic stages have their dynamics driven by sea temperature during the planktonic stages, while the shorter planktonic-stage species or species with no planktonic stage are driven by aragonite saturation state. The work certainly warrants publishing however I think further analysis in relation to other important environment variables should be included to elucidate the response to aragonite saturation state which is clearly driven (and this is mentioned in this paper) by a multitude of other factors, as well as some more discussion of the datasets. I detail this in comments below.

Major comments:

Correlation to aragonite saturation state – there are two issues here:

1. From what I understand, the carbonate data used in this study (pH and aragonite) were taken from a published study (Salisbury and Jonsson 2018), which are calculated means for the whole of the Gulf of Maine. This is very different to the temperature data, which was taken from buoys closest to the Island, and then using only the buoy that had the most amount of data. While I appreciate it is difficult to get high quality carbonate data compared to temperature data, the authors need to acknowledge the caveats associated with: Firstly using calculated variables; secondly, the difference in what is considered as the annual cycle (e.g. whether it makes a difference to have annual averages for each year (Dec – Dec) or the preceding year average (summer to summer) – this can be tested using the temperature data); and thirdly, using variables that are averaged over an area that covers over 4 degrees of longitude! Given the proximity to land, and the fact these are rocky shore species, the variability in carbonate parameters that these rocky shore species experience is likely to be different to the larger regional annual average. As I said, I appreciate it is difficult to get data – however this should not preclude this from being discussed; including literature that has shown just how variable the near shore can be (e.g. Wootton et al. 2008) and the impacts of variability on near shore species (e.g. Mangan et al. 2017 and references therein for other studies of nearshore pH variability).

2. Connected to the previous - Salisbury and Jonsson (2018) discuss the importance of salinity as an influencing factor on the carbonate system in this region. More analysis is required to see if this can be elucidated any further. Especially as the authors in this manuscript state “Notably, Ω_{ar} in the Gulf of Maine co-varies with acidification, warming and salinity... so we hypothesize Ω_{ar} may be a proxy for the combined and adverse effects of acidification, warming and salinity” (P5, line83-86). Salinity data should be easier to come by compared to carbonate data, and so it should not be too difficult to analyse whether salinity itself is correlated with any of the biological observations. Using aragonite as a proxy for multiple factors does not really help us to understand drivers of declining species. The fact that pH does not come up with a significant correlation suggests that acidification (at least using this annual average data) is unlikely to be a driving factor – or at least is being masked by the effects of temperature and salinity (as per Salisbury and Jonsson (2018)’s discussion). Given the declines in abundance correlate with higher levels of saturation, which is counter intuitive for a purely acidifying ocean, it seems likely that the other drivers of aragonite (salinity being a significant one via its impact on alkalinity) will produce a stronger correlation.

Minor comments:

P2 L16: "led" instead of "lead"

P4 L65-66: "...can disrupt calcification of molluscan shells" – while this is true, it has been shown that acidification primarily operates by making it more energetically costly to build/repair shells, and hence a shift in energy budget might occur that allows shells to continue to be build. There are also a whole range of physiological processes that could be impacted by elevated pCO₂, lowered pH (e.g. Portner et al. 2004). This also means that any change in food may also affect the energy budget and either overcome or facilitate the impacts from acidification. This is especially important when considering early life stages that may not yet calcify (e.g. barnacle larvae), but early life stages have been shown to be likely to be most vulnerable. This should be discussed.

P4 L68: "are related" should be replaced with "are correlated". The analysis in this paper is purely correlation – there is no definite "cause and effect" link established here, this should be made clear throughout.

P5 L82-83: See second minor comment above - Lower rates of calcification do not necessarily equate to lower abundances. Organisms could still be growing and present, but there may be impacts that have not been recorded/observed. For example, although there was larger abundance, were the individuals smaller (confirming an impact of OA on growth)? This also relates to my major comments about whether salinity is the real driver here rather than aragonite.

P4 L90-94: I do not find it surprising that declines are correlated – you will get a correlation between any species that is declining over the same period! I don't see the value of this being put in the main manuscript – perhaps remove to supplementary. Unless there is some ecosystem link that is worthy of discussing in the main paper.

P6 L107: barnacles do not begin calcification as a pelagic larvae, but they do during settlement and transformation from cyprid to juvenile (and so is captured by the measure of recruitment here); also the combination of warming and acidification has been shown to have a greater impact on recruitment than acidification alone (Findlay et al. 2009). This should be discussed.

P7 L129-139: It would be nice to see the possibility of replacement species (and/or examples of this) being discussed in this paragraph – rather than just assuming something will decline and nothing will replace it there may be similar species that do the same job that can replace them (e.g. *Elminius modestus* replaces *Semibalanus balanoides*).

P9 L189: remove word "from" in "two sites were dropped from because..."

P10 L199-212: Given the lack of data for carbonate parameters (and major comment above) it would be good to see analysis of what difference it makes using annual averages compared to monthly. And also, what difference it makes using July to June annual data compared to January to December data. The carbonate chemistry of the water is going to be associated with and impacted by temperature, and so it seems wrong to treat the environmental data differently. I'd like to see this explored more, even if it is only presented in supplementary but discussed in the main manuscript.

Figure: I would like to see the environmental data plotted up as a time-series, this could be added to the top of Figure 1 (or a separate figure). With temperature, pH and aragonite annual averages plotted against year. The authors should also use the correct name for the average they used – are the averages in figure 1 means? If so, please use the term mean.

References:

Wootton et al. 2008. Dynamic patterns and ecological impacts of declining ocean pH in a high-resolution multi-year dataset. PNAS, 105: 18848–18853, doi:10.1073/pnas.0810079105
Mangan et al. 2017. Fluctuating seawater pH/pCO₂ regimes are more energetically expensive than static pH/pCO₂ levels in the mussel *Mytilus edulis*. Proc. R. Soc. B 284: 20171642, doi.org/10.1098/rspb.2017.1642
Portner et al. 2004. Biological Impact of Elevated Ocean CO₂ Concentrations: Lessons from Animal Physiology and Earth History. Journal of Oceanography. 60: 705-718.
Findlay et al. 2009. Post-larval development of two intertidal barnacles at elevated CO₂ and temperature. Mar Biol. DOI:10.1007/s00227-009-1356-1

Reviewer #2 (Remarks to the Author):

Petratis and Dudgeon provide a manuscript which highlights the long-term changes in the abundance and recruitment of five ecologically important intertidal species due to anthropogenic changes in the oceanic conditions. The authors suggest which environmental conditions are most likely to be driving such changes; based on the life-history characteristics of the species and the most parsimonious statistical model. Finally, the authors emphasise the likely ecosystem consequences for such a species decline into the future in terms of further changes in community composition/food-web and functioning.

At its core, this work is classic ecology with a relatively simple (but long-term) approach. However, this should not be taken as a bad thing, as I agree with the authors that such long-term datasets are uncommon and yet incredibly useful in predicting the effects of climate change. Most predictions considering the long-term implications climate change are reliant on either latitudinal gradients, natural analogues or retrospective surveys, and thus have to utilise some form of inference. As a note, I was particularly glad to see the results being linked to changes in both warming and acidification, as I so often see research falling into attributions of just one or the other; whereas the response of specific species are highly likely to be dictated by one, the other, or their combination.

The manuscript is concise and very much to the point, and I found the work to be convincing in the evidence provided. The statistics appear to be technically sound and valid, with the requisite information provided in the supplementary data. I believe that it should be easily possible to reproduce the work here, however, although the authors do highlight that the raw data is available on request – I feel that it may be far more convenient to provide the data along with the supplementary data or within a suitable repository.

I feel that the manuscript will be of interest to those in the community and the wider field by providing actual measured changes in the community over time (rather than an inference). As the work is very much in line with current thinking, I do not believe that it will fundamentally challenge the way in which people within the field will think, but I do believe that it will provide highly useful evidence and data required to begin pushing the field further forward.

In summary, the manuscript was interesting and a pleasure to read, and for the first time ever I actually have no specific criticisms to provide.

Ben Harvey
Shimoda Marine Research Center, University of Tsukuba

Reviewer #3 (Remarks to the Author):

This is a very interesting paper that broke the trance-like state of wondering where and when one might next source loo-roll...and fuelling frustration that lock-down means that we are not continuing our own long-term observations stretching back 70 years (with gaps) on some really good tides this week.

These are interesting and novel observations, taking advantage of the controls from long-term manipulative experiments stretching back 20 years enabling sustained observing of trends. The recording of species declining in the face of climate change are particularly important and rare, as are the discussions on the consequences for community structure and functioning. Most of literature is about advancing species and Parmesan in an early meta analysis noted that fewer retreats have been observed; you have done this and also for the all the key species in an already low diversity community.

I have no major concerns about this work which deserves a wide audience and is generally very sound. I do, however, have a few queries and suggestions - some of which address consequences of this work being published in a short-format journal which leads to inevitable glossing over of detail to get over the bigger picture in a short space.

General Comments

1) I am very surprised there is no mention of the long-term research showing responses to climate fluctuations and more recent rapid climate change (since around 1989) in the N E Atlantic (Line 28), including much work on rocky shores by Southward (1991 JMBA), Hawkins, Burrows (Burrows et al 2020 GCB), Mieszkowska and co-workers (see Poloczanska et al 2008 Ecology; Hawkins et al 2003 Sci Total Envir, 2008 Climate Res, 2009 MEPS; Helmuth et al 2006 AREES, Philippartt et al 2011, JEMBE). Back in the 1950s Southward and Crisp (1954 J.anim Ecol) showed that *Semibalanus* declined as it got warmer, with Southward subsequently recovering when it got cold again in the 1960s, before doing less well (but still persisting in the British Isles) when it got warmer (Southward et al 1995 J.Therm Biol, Mieszkowska et al 2014 J.mar-systems). Wetthey and colleagues showed it disappeared in Spain and have recently come up with mechanistic explanations for its decline on both sides of the Atlantic (papers in JMBA and JEMBE on reproductive failure in warm years). Is there any evidence of warm Decembers messing up the onset of breeding??

Similar changes have been recorded in Europe in cold water limpets such as *Patella vulgata* (Southward et al 1995 and Hawkins et al 2008) with recent warming, and *Tectura testudinalis* has disappeared from the shores around the Isle of Man at its southern range edge in Europe (mentioned in Hawkins et al. 2009).

2) The above is pertinent as in several places the ms talks about changes in the N Atlantic, whereas really this should state NW Atlantic. The authors could point out the differences between the two sides (see Jenkins et al 2008 Ecology). Due to glacial recolonisation processes most of the shore biota in the Gulf of Maine are northern species that probably recolonised from Europe via Iceland and Greenland. There are no southern species to take the place of those declining, in contrast to the NE Atlantic where similar warm water species are advancing or increasing and taking the place of cold water functional equivalents (*P.depressa* replacing *P.vulgata*, *Chthamalus* spp replacing *Semibalanus*, trochids replacing *L.littorea*). Thus the food webs of New England/Canada are particularly vulnerable to disruption as there are no warm water substitutes to come and fill a void. With the barnacles, the local Chthamalid is probably unable to get around Cape Cod to fill any gaps and there are no trochids or patellids. This is an exciting extra point to emphasize.(see above listed work for references - mostly in Hawkins et al 2019 Chapter 2 in Interactions in the Marine Benthos, CUP)

3) How typical is the study area of the rest of the coast? This needs emphasizing with evidence.

4) Do you have any data pre-1997 for this coastal area that could even qualitatively stretch back

further?

5) There is a very exciting strong peak around 2006 in most of the focal species - and whilst the overall trend is downward, this does show the capability of the system to bounce back. Do you have any ideas what caused this? It should certainly be discussed somewhere. If the dots were joined up it would be a fluctuating curve and the emphasis on % declines per year suggests a monotonic trend which is not the case. *Semibalanus* in particular typically has very big recruitment years where there is very good match of larval release and the phytoplankton bloom (first spotted by Connell 1961 in *Ecological monographs*; see also Hawkins and Hartnoll, 1982 *JEMBE*). Such peaks in recruitment are not unknown in *Mytilus*. Only one good recruitment in a lifespan (once every 5-6 years??) is sufficient for persistence. The concordance in all these different species suggests some kind of mesoscale or local scale driver - any ideas?

6) What are the consequences of recruitment for adult cover of *Mytilus* and *Semibalanus* - is there any density dependent feedback as shown by Jenkins et al 2008 (*J. anim. ecol.*). Could these data be added?

7) Figure 1 is a very dense figure and would benefit from labelling of axes (what units), some labels on graphs (nos, recruitment etc) and referring to different letters for boxes in the legends. Is March the winter minimum temperature - say so in legend?

8) Climate strictly refers to the the physical regime of temperature, precipitation, winds etc. Ocean acidification is a co-variate of climate change as both are driven by greenhouse gas Carbon dioxide - but strictly it is a change in biogeochemical processes - you can lump both as global change. This also begs the question is aragonite saturation driving these changes in ecology or just reflecting climate change and co-varying with it (temperature and solubility for instance?). You do say it is general proxy - bit more clarification or qualification would help, perhaps?

9) *L. obtusata* lives in a damp refuge - canopy algae - might this explain the lack of trend due to facilitation providing a refuge. In this worth commenting on? (see Moore et al 2007 *MEPS* for work on Northern limpets under fucoids)?

10) Fucoids as cold water species may also die as they are susceptible to bleaching events in warm years as adults and also when recruiting. The die off of grazers may enable persistence, as would die-off of *Nucella* as denser barnacles facilitate recruitment of *Fucus*. The interesting thing is the aphasia of these responses to climate. There is some evidence of *Fucus* declines at their southern limits in Europe due to both greater physical stress and more grazing from all those extra clades (*patella*, *trochids*) missing in New England (Ferriera et al 2015 *MEPS*).

Specific comments by line number

28 - other refs? These refs are mainly about OA - not climate

54 - is the decline linear - or a line through a wobble - is % decline per year a valid metric?

76-78 Any links to stratification of Gulf of Maine and ultimately NAO?? Windier late winters could mean messier and less strong phytoplankton and thus less good match with larval release (there is good evidence for this in *Semibalanus*). Less stratification could also mean weaker fronts retaining larvae??

94 - any correlation (with lags?) of *Nucella* with barnacles and mussels??

110 swathes

124 - although the grazing regime might get back to pre-1860, the system as a whole will not due

to all the other climate driven changes. This is an important point as this is a non-native driven system from the top include Green(shore) crabs. Perhaps just rephrase little.

306 typo - author missing in reference???

Summary

Very nice work!

Steve Hawkins

Responses to reviewers' comments

We would like to thank the reviewers for their thoughtful and extensive comments. Before addressing specific comments from each reviewer, we have several general comments that are relevant. First, the reviewers' comments are numbered by reviewer and comment; for example, the first comment by reviewer 3 is numbered 3.1. Second, many of the comments are requests about more detail, but we were very constrained by the word limit of Nature Ecology & Evolution, where the manuscript was originally submitted before being transferred to Communications Biology. The text for a Brief Communications to Nature Ecology & Evolution has typically 1,000-1,500 words, excluding abstract, references and figure legends. Our original submission was just under 1500 words and while we were aware of many of the issues raised by the reviewers, we could not fully address them given the word limit. Third, we have included as much new information as we could without substantially altering the focus and intent of the original manuscript. To fully address the reviewers' comments would have entailed writing a very different manuscript rather than a revision. We hope we were able to address as fully as possible the reviewers' comments in the revision while retaining the scope and aims of the original manuscript.

Reviewers' comments:

Reviewer #1: Biological oceanography & marine species effects of climate change and ocean acidification

This paper details the decline in recruitment and abundance of 5 key rocky shore species as a result of climate change, specifically temperature and ocean acidification. The analysis is novel, although some of the data has been published in other papers (both part of the biological and the environmental data). It is an interesting study and nice to see biological observations and monitoring being considered in the context of climate change and alongside the more routine observations of the physico-chemical environment. The authors propose that the species with longer planktonic stages have their dynamics driven by sea temperature during the planktonic stages, while the shorter planktonic-stage species or species with no planktonic stage are driven by aragonite saturation state. The work certainly warrants publishing however I think further analysis in relation to other important environment variables should be included to elucidate the response to aragonite saturation state which is clearly driven (and this is mentioned in this paper) by a multitude of other factors, as well as some more discussion of the datasets. I detail this in comments below.

Major comments:

Correlation to aragonite saturation state – there are two issues here:

1.1 From what I understand, the carbonate data used in this study (pH and aragonite) were taken from a published study (Salisbury and Jonsson 2018), which are calculated means for the whole of the Gulf of Maine. This is very different to the temperature data, which was taken from buoys closest to the Island, and then using only the buoy that had

the most amount of data. While I appreciate it is difficult to get high quality carbonate data compared to temperature data, the authors need to acknowledge the caveats associated with: Firstly using calculated variables; secondly, the difference in what is considered as the annual cycle (e.g. whether it makes a difference to have annual averages for each year (Dec – Dec) or the preceding year average (summer to summer) – this can be tested using the temperature data); and thirdly, using variables that are averaged over an area that covers over 4 degrees of longitude! Given the proximity to land, and the fact these are rocky shore species, the variability in carbonate parameters that these rocky shore species experience is likely to be different to the larger regional annual average.

These are all very good suggestions and we acknowledge some of the reviewer's concerns on lines 99-102 and 164-175. Lines 99-102 read, "Notably, Ω_{AR} in the Gulf of Maine co-varies with not only acidification, warming and salinity but also local nearshore conditions²¹, and so we hypothesize Ω_{AR} may be a proxy for the combined and adverse effects of acidification, warming and salinity."

Lines 164-175 read, "The Gulf of Maine has also seen declines in pH and aragonite saturation state (Ω_{AR}) in the Gulf of Maine; from 1981 to 2014, and the rates of decline for pH and Ω_{AR} have been 0.0018 year^{-1} and 0.049 year^{-1} , respectively^{12,21}. However, our observations of the links between the changes in water chemistry and declines in species abundances should be viewed with caution for several reasons. The values for pH and Ω_{AR} , which were extracted from the literature, are average yearly estimates across the entire Gulf of Maine. In the Gulf of Maine, values for pH and, in particular, Ω_{AR} show large annual and decadal variation and covary with changes in salinity, total alkalinity and water temperature²¹."

Note that the superscripted references 12 and 21 are Wang et al. (2013) and Salisbury and Jönsson (2018), respectively.

1.2...As I said, I appreciate it is difficult to get data – however this should not preclude this from being discussed; including literature that has shown just how variable the near shore can be (e.g. Wootton et al. 2008) and the impacts of variability on near shore species (e.g. Mangan et al. 2017 and references therein for other studies of nearshore pH variability).

We cite Wootton et al. (2008) in the original ms and in the revision on line 33 as an example of one of the very few long-term studies. Wootton et al. (2008) do discuss variation in pH but their data are from a single site in the Pacific that was sampled over eight years. Thus, the data are not spatially replicated and so there is no way of knowing if the temporal variation at that single site is an unbiased estimate of average temporal variation among sites. The single site used by Wootton et al. could be an outlier. Salisbury and Jönsson (2018) is a relevant citation for diurnal and annual variation in pH because their data not only cover a longer time period and over a larger number of

sampling locations but also cover the time period and location in the Gulf of Maine of our data and analyses.

*Mangan et al. (2017) is lab study of the physiological effects of changes in pH on adult mussels (*Mytilus edulis*), and we are not sure how this particular study is relevant to our long-term field study of recruitment and abundance of not only mussels but also an additional five species. We think the better and more comprehensive citations are: Kroeker et al. (2013), Preslajski et al. (2015), and Waldbusser and Salisbury (2014), which in the ms, are superscripted citations 1,3 and 4, respectively. Waldbusser and Salisbury (2014), in particular, is an exceptional review of physiological effects.*

1.3. Connected to the previous - Salisbury and Jonsson (2018) discuss the importance of salinity as an influencing factor on the carbonate system in this region. More analysis is required to see if this can be elucidated any further. Especially as the authors in this manuscript state “Notably, Ω_{ar} in the Gulf of Maine co-varies with acidification, warming and salinity... so we hypothesize Ω_{ar} may be a proxy for the combined and adverse effects of acidification, warming and salinity” (P5, line83-86). Salinity data should be easier to come by compared to carbonate data, and so it should not be too difficult to analyse whether salinity itself is correlated with any of the biological observations. Using aragonite as a proxy for multiple factors does not really help us to understand drivers of declining species. The fact that pH does not come up with a significant correlation suggests that acidification (at least using this annual average data) is unlikely to be a driving factor – or at least is being masked by the effects of temperature and salinity (as per Salisbury and Jonsson (2018)’s discussion). Given the declines in abundance correlate with higher levels of saturation, which is counter intuitive for a purely acidifying ocean, it seems likely that the other drivers of aragonite (salinity being a significant one via its impact on alkalinity) will produce a stronger correlation.

The reviewer suggests a logical sequence of an observation and an inference: 1) aragonite saturation ratio is strongly affected by salinity and 2) therefore salinity might be either a better predictor of our observed declines or a confounding effect that needs to be included in the models. We agree with the observation, and Salisbury and Jönsson (2018) fully discuss this point. Lines 170-172 now read, “In the Gulf of Maine, values for pH and, in particular, Ω_{AR} show large annual and decadal variation and covary with changes in salinity, total alkalinity and water temperature²¹.”

However, we think the inference cannot be successfully examined because of the lack of available data on salinity at the appropriate scale and given the physiological characteristics of the species we examined. The only long-term data for salinity available are from off-shore buoys and off-shore salinities do not vary too much. The long-term average salinity from 1984 to 2015 in the Gulf of Maine ranges from 31 to 33.5 (Salisbury and Jönsson 2018, their Fig. 3). In addition, nearshore salinities in the Gulf of Maine are strongly affected and quite variable due to freshwater run-off and inputs from streams and rivers (we note this point on lines 172-173). More to the point, one of us (Dudgeon) collected salinity data in the late 1980s Frenchman’s Bay, Maine, which is approximately 35 km from our sites. During spring run-off, salinities were

typically around 26 along the coast. All in all, it is highly unlikely that measurements of off-shore salinity would be an accurate predictor of nearshore salinity. Second, the species examined in our study are euryhaline and can easily adapt to short-term and long-term changes in salinity. Taken together, we think it is very unlikely offshore data for salinity would show a correlation with changes in abundance and recruitment.

We note that the same objections about using offshore data to predict nearshore conditions also applies to temperature, pH and Ω_{AR} , and these issues were raised by Reviewer 1. We find it a bit disingenuous that Reviewer 1 is now overlooking his/her concerns about temperature, pH and Ω_{AR} and advocating for doing the same analysis using salinity.

Finally, we note that our raw data are available online and available for others to data-mine as they wish.

1.4. P2 L16: “led” instead of “lead” *Corrected*

1.5. P4 L65-66: “...can disrupt calcification of molluscan shells” – while this is true, it has been shown that acidification primarily operates by making it more energetically costly to build/repair shells, and hence a shift in energy budget might occur that allows shells to continue to be build. There are also a whole range of physiological processes that could be impacted by elevated pCO₂, lowered pH (e.g. Portner et al. 2004). This also means that any change in food may also affect the energy budget and either overcome or facilitate the impacts from acidification. This is especially important when considering early life stages that may not yet calcify (e.g. barnacle larvae), but early life stages have been shown to be likely to be most vulnerable. This should be discussed.

Portner et al. (2004) is a relatively old review, and we think Waldbusser and Salisbury (2014), which we cite in both the original and revised manuscripts, provides a more up-to-date review of physiological effects. In addition, in the first half of the reviewer’s comment, it seems he/she objects to the use of the word, “disrupt.” We have changed it to “slow.” Lines 41-44 read, “Changes in water temperature, pH and Ω_{AR} affect the development, growth, calcification and survivorship of marine organisms^{1,3}, and in particular, changes in ocean pH and Ω_{AR} can slow calcification of molluscan shells⁴.”

Superscripted references 1, 3 and 4 are Kroeker et al. (2013), Przeslawski et al. (2015) and Waldbusser and Salisbury (2014), respectively.

1.6. P4 L68: “are related” should be replaced with “are correlated”. The analysis in this paper is purely correlation – there is no definite “cause and effect” link established here, this should be made clear throughout.

We have made this change and additional changes throughout the ms to emphasize we have not direct evidence for causal links.

1.7. P5 L82-83: See second minor comment above - Lower rates of calcification do not necessarily equate to lower abundances. Organisms could still be growing and present, but there may be impacts that have not been recorded/observed. For example, although there was larger abundance, were the individuals smaller (confirming an impact of OA on growth)? This also relates to my major comments about whether salinity is the real driver here rather than aragonite.

Please see replies to comments 1.1, 1.3, and 1.5.

1.8. P4 L90-94: I do not find it surprising that declines are correlated – you will get a correlation between any species that is declining over the same period! I don't see the value of this being put in the main manuscript – perhaps remove to supplementary. Unless there is some ecosystem link that is worthy of discussing in the main paper.

The reviewer is correct that declines would be correlated because they are, by definition, all declining over the same time period. We have re-done the correlations using residuals from each curve, and found the residuals are correlated. We think the correlations suggest that local, small-scale differences affect the abundances of gastropods in a similar fashion. We have replaced Supplemental Table 5 with a new Supplemental Table 8 with the correlations among the residuals.

Lines 103-109 read, “Two additional observations suggest local, small-scale characteristics, such as the aspect of the shore and tidal flow within specific bays, also play a role. First is the large amount of spatial variability. Between 20% and 50% of the total variance in abundances and recruitment is due to variation among sites (Supplemental Table 7). Second, the residuals from the predicted declines are correlated (Supplemental Table 8). Four of the six pair-wise correlations of abundances among the gastropods are significant and range from +0.154 to +0.418 ($p < 0.01$ in all cases).”

1.9. P6 L107: barnacles do not begin calcification as a pelagic larvae, but they do during settlement and transformation from cyprid to juvenile (and so is captured by the measure of recruitment here); also the combination of warming and acidification has been shown to have a greater impact on recruitment than acidification alone (Findlay et al. 2009). This should be discussed.

*It is not clear to us how Findlay et al. (2009) is relevant to our data and results on the decline in barnacle recruitment. Findlay et al. (2009) used the same species for which we have recruitment data (*S. balanoides*), but the study is a lab experiment of the effects of high CO₂ on adults and embryonic development. The high CO₂ treatment reduced pH to 7.7, which is well outside the range of pH reported by Salisbury and Jönsson (2018) for the Gulf of Maine. In addition, slower embryonic development as reported by Findlay et al. (2009) does not necessarily translate into a reduction in barnacle recruitment, which is what we sampled.*

In addition, our analysis of model selection for barnacle recruitment examined all possible models with one or two independent variables, and none of the top seven models

included both pH and temperature (see Supplemental Table 5). The intercept only model was best supported and far better supported than any other model.

1.10 P7 L129-139: It would be nice to see the possibility of replacement species (and/or examples of this) being discussed in this paragraph – rather than just assuming something will decline and nothing will replace it there may be similar species that do the same job that can replace them (e.g. *Elminius modestus* replaces *Semibalanus balanoides*).

This is a very interesting point and was also noted by Steve Hawkins (Reviewer 3, see comment 3.2). See lines 189-195, which read, “Unlike what has been predicted for the shores of Great Britain and western Europe, it is unlikely that southern species will expand their ranges northward to fill the gaps^{10,32}. Recolonization of rocky intertidal shores in the Gulf of Maine after the last glacial retreat was by European species rather than range expansions of southern species^{33,34}.”

Superscripted references 10, 32, 33, and 34 are Burrows et al. (2020), Wetthey et al. (2011), Maggs et al. (2008), and Wares and Cunningham (2001), respectively.

1.11. P9 L189: remove word “from” in “two sites were dropped from because...”

Done

1.12. P10 L199-212: Given the lack of data for carbonate parameters (and major comment above) it would be good to see analysis of what difference it makes using annual averages compared to monthly. And also, what difference it makes using July to June annual data compared to January to December data. The carbonate chemistry of the water is going to be associated with and impacted by temperature, and so it seems wrong to treat the environmental data differently. I’d like to see this explored more, even if it is only presented in supplementary but discussed in the main manuscript.

*We do not understand this comment. We do not have monthly averages for pH or Ω_{AR} ; we only have Jan-Dec averages for the yearly averages. We have monthly averages for temperature. For mussel and barnacle recruitment, we used temperature data for the months for which larvae of mussels and barnacles were likely to be in the water column (e.g. June, July and August for mussels). For the abundance data, which were usually collected in July, we used the yearly July-July averages for temperature. To handle the fact that we had only Jan-Dec averages for carbonate parameters, we included the current year and the previous year as independent variables for selection of the best model for both recruitment and abundance (see Supplemental Table 4). Clearly this adds unwanted noise for using the carbonate data, but even so, the best or second-best models for *L. obtusata*, *N. lapillus*, and *T. testudinalis* involve Ω_{AR} . We have added some discussion of this problem (see our reply to comment 1.1).*

1.13. Figure: I would like to see the environmental data plotted up as a time-series, this could be added to the top of Figure 1 (or a separate figure). With temperature, pH and aragonite annual averages plotted against year. The authors should also use the correct

name for the average they used – are the averages in figure 1 means? If so, please use the term mean.

We have added a Figure of the time series for temperature (see new Supplemental Figure 1). We are reluctant to plot annual averages for pH and Ω_{AR} because these plots are given as Figure 7 in Salisbury and Jönsson (2018). We do not wish to clutter our paper with Figures that already exist in the literature. Moreover, in order to re-plot time-series of these variables, we would have to obtain copyright permission.

We have changed averages to mean and made clear that the use of mean refers to the sample mean. We note, however, that the senior author (Petraitis) co-taught advanced statistics for over 20 years with Prof Warren Ewens, FRS, who is a highly regarded theoretical statistician. Petraitis was often gently chided by Ewens that one should not use the term “mean,” which is a parameter, for the “sample mean.” Among statisticians, the correct term for the estimate of the mean or the sample mean is “average.” Old habits are hard to change, but we have replaced “average” with “mean.” throughout.

References:

- Wootton et al. 2008. Dynamic patterns and ecological impacts of declining ocean pH in a high-resolution multi-year dataset. PNAS, 105: 18848–18853, doi:10.1073/pnas.0810079105
- Mangan et al. 2017. Fluctuating seawater pH/pCO₂ regimes are more energetically expensive than static pH/pCO₂ levels in the mussel *Mytilus edulis*. Proc. R. Soc. B 284: 20171642, doi.org/10.1098/rspb.2017.1642
- Portner et al. 2004. Biological Impact of Elevated Ocean CO₂ Concentrations: Lessons from Animal Physiology and Earth History. Journal of Oceanography. 60: 705-718.
- Findlay et al. 2009. Post-larval development of two intertidal barnacles at elevated CO₂ and temperature. Mar Biol. DOI:10.1007/s00227-009-1356-1

Reviewer #2: Marine climate change ecology (Ben Harvey, Shimoda Marine Research Center, University of Tsukuba)

Petraitis and Dudgeon provide a manuscript which highlights the long-term changes in the abundance and recruitment of five ecologically important intertidal species due to anthropogenic changes in the oceanic conditions. The authors suggest which environmental conditions are most likely to be driving such changes; based on the life-history characteristics of the species and the most parsimonious statistical model. Finally, the authors emphasise the likely ecosystem consequences for such a species decline into the future in terms of further changes in community composition/food-web and functioning.

At its core, this work is classic ecology with a relatively simple (but long-term) approach. However, this should not be taken as a bad thing, as I agree with the authors that such long-term datasets are uncommon and yet incredibly useful in predicting the effects of climate change. Most predictions considering the long-term implications climate change are reliant on either latitudinal gradients, natural analogues or retrospective surveys, and

thus have to utilise some form of inference. As a note, I was particularly glad to see the results being linked to changes in both warming and acidification, as I so often see research falling into attributions of just one or the other; whereas the response of specific species are highly likely to be dictated by one, the other, or their combination.

The manuscript is concise and very much to the point, and I found the work to be convincing in the evidence provided. The statistics appear to be technically sound and valid, with the requisite information provided in the supplementary data. I believe that it should be easily possible to reproduce the work here, however, although the authors do highlight that the raw data is available on request – I feel that it may be far more convenient to provide the data along with the supplementary data or within a suitable repository.

I feel that the manuscript will be of interest to those in the community and the wider field by providing actual measured changes in the community over time (rather than an inference). As the work is very much in line with current thinking, I do not believe that it will fundamentally challenge the way in which people within the field will think, but I do believe that it will provide highly useful evidence and data required to begin pushing the field further forward.

In summary, the manuscript was interesting and a pleasure to read, and for the first time ever I actually have no specific criticisms to provide.

We thank Dr. Harvey, Reviewer 2, for his very kind comments.

Reviewer #3: Climate change effects on intertidal ecology (Steve Hawkins)

This is a very interesting paper that broke the trance-like state of wondering where and when one might next source loo-roll...and fuelling frustration that lock-down means that we are not continuing our own long-term observations stretching back 70 years (with gaps) on some really good tides this week.

These are interesting and novel observations, taking advantage of the controls from long-term manipulative experiments stretching back 20 years enabling sustained observing of trends. The recording of species declining in the face of climate change are particularly important and rare, as are the discussions on the consequences for community structure and functioning. Most of literature is about advancing species and Parmesan in an early meta analysis noted that fewer retreats have been observed; you have done this and also for the all the key species in an already low diversity community.

I have no major concerns about this work, which deserves a wide audience and is generally very sound. I do, however, have a few queries and suggestions - some of which address consequences of this work being published in a short-format journal which leads to inevitable glossing over of detail to get over the bigger picture in a short space.

3.1. I am very surprised there is no mention of the long-term research showing responses to climate fluctuations and more recent rapid climate change (since around 1989) in the N E Atlantic (Line 28), including much work on rocky shores by Southward (1991 JMBA), Hawkins, Burrows (Burrows et al 2020 GCB), Mieszkowska and co-workers (see Poloczanska et al 2008 Ecology; Hawkins et al 2003 Sci Total Envir, 2008 Climate Res, 2009 MEPS; Helmuth et al 2006 AREES, Philippart et al 2011, JEMBE). Back in the 1950s Southward and Crisp (1954 J.anim Ecol) showed that *Semibalanus* declined as it got warmer, with Southward subsequently recovering when it got cold again in the 1960s, before doing less well (but still persisting in the British Isles) when it got warmer (Southward et al 1995 J.Therm Biol, Mieszkowska et al 2014 J.mar-systems). Wetthey and colleagues showed it disappeared in Spain and have recently come up with mechanistic explanations for its decline on both sides of the Atlantic (papers in JMBA and JEMBE on reproductive failure in warm years). Is there any evidence of warm Decembers messing up the onset of breeding?? Similar changes have been recorded in Europe in cold water limpets such as *Patella vulgata* (Southward et al 1995 and Hawkins et al 2008) with recent warming, and *Tectura testudinalis* has disappeared from the shores around the Isle of Man at its southern range edge in Europe (mentioned in Hawkins et al. 2009).

We appreciate Reviewer 3's suggested references, which he thinks might be relevant, but we think, if anything, most of the papers provide excellent examples of how many climate predictions for marine invertebrates are based on very short-term surveys and/or modeling. We made this point in the introductory comments in our original submission and in the revision (lines 28-30) and adding one of his suggested references (superscripted reference 5, which is Helmuth et al., 2006).

*We also include here a brief commentary on the papers, which we do not discuss in detail in our revised manuscript. The single paper that contains original long-term data is Southward (1991). Southward provides annual data on *Semibalanus balanoides* abundance from 1951 to 1990 at several sites. This is an exceptional data set, which we cite in our revised ms. Note however, these are data on abundance and we analyzed recruitment. Also note that Hawkins et al. (2003, 2008, 2009) and Mieszkowska et al. (2014) used Southward's (1991) data.*

*We included comments about Southward (1991) in the revision, and lines 120-123 read, "Southward's data on abundance of *S. balanoides* and other barnacle species from 1951 to 1990 are the only other comparable data from the north Atlantic Ocean²². While this remarkable set of data has been expanded by other researchers and used in a number of studies, none report a decline in barnacle recruitment or abundances^{5,23-25}."*

Superscripted references 5, 22, 23, 24, and 25 are Helmuth et al. (2006), Southward (1991), Hawkins et al. (2003), Hawkins et al. (2009), and Mieszkowska et al. (2014), respectively.

Commentary on suggested references.

*Burrows et al. (2020): The primary focus of this paper is changes in species composition due to warming. The authors estimate thermal limits of species and then form an aggregate measure to examine changes in species composition due to warming. Data are from long-term sampling of some sites for 17 years (2002–2018) and other sites for 41 years (1976–2018). Abundance data were binned into abundance categories. Authors do not cite or provide access to raw data or the aggregated data. Rates of change discussed in the paper are not comparable to our data because of the authors' use of categories rather than actual counts of abundance. Even so, there is some evidence for declines in abundance of *M. edulis* and *N. lapillus*, which is consistent with our results. However, the authors' results also show increases in *L. littorea* and *S. balanoides*, which are not consistent with our results.*

*Hawkins et al. (2008): This is a review of changes in abundances in Great Britain without formal statistical analyses. The authors mention *S. balanoides* and *T. testudinalis*, but no other species relevant to our paper. Figure 2 shows a plot of *S. balanoides* abundance using Southward's (1991) data with some additional data starting in the late 1990s and going through 2004.*

*Hawkins et al. (2003): This paper uses Southward's (1991) data from 1950 to 2001 for *S. balanoides* abundance and shows a weak negative correlation with annual SST based large area (5 degree square).*

*Hawkins et al. (2009): This is a review paper that discusses the retreat and decline of *S. balanoides* in the N.E. Atlantic Ocean. The paper includes a model based on Southward's (1991) data.*

*Helmuth et al. (2006): Review paper that briefly mentions effects of climate change on *M. edulis* and *T. testudinalis*.*

*Jones et al. (2012): This paper examines changes in range and abundance of *S. balanoides* in the N.W. Atlantic Ocean from North Carolina to Cape Cod. Data are from sampling of 8 sites in 1963 and 14 in 2007 and uses abundance categories. There is no formal analysis. The 1963 data was extracted from field notes of Southward*

*Mieszkowska et al. (2014): This is a data mining paper in which the correlation of *S. balanoides* abundance from 1953 to 2008 with Atlantic Multidecadal Oscillation (AMO) and NAO was studied, and included lags. Abundance data are from Southward (1991) and other sources that continued sampling Southward's sites. Authors found a weak correlation with AMO and no correlation with NAO.*

Philippart et al. (2011): A review of changes in oceanic conditions throughout Europe with very little discussion of intertidal species.

*Poloczanska et al. (2008): This is a modeling paper. Recruitment of *S. balanoides* was inferred; using Southward's (1991) data.*

Southward (1991): Abundance of S. balanoides from 1951 to 1990 and source of data of many other papers from Hawkins and his collaborators.

Southward and Crisp (1954): This paper is based on four years of data on settlement of S. balanoides. There is no formal analysis and the authors suggest warm air and sea temperatures “may” be responsible for decline in settlement.

Southward et al. (1995): This is a review paper that mentions changes in the ranges of L. littorea and N. lapillus and contains some information on abundance of S. balanoides.

Wethey et al. (2011): This is a modeling paper of predicted changes in SST and species abundances in Europe and Great Britain. Predictions were “tested” by sampling over a few years to examine if changes in abundances were consistent with the predictions. No formal statistical test of observed changes against predicted changes.

References

- Burrows et al. 2020. Global Change Biology 26:2093-2105.*
Hawkins et al. 2008. Climate Research 37:123-133.
Hawkins et al. 2003. Science of the Total Environment 310:245-256.
Hawkins et al. 2009. Marine Ecology Progress Series 396:245-259.
Helmuth et al. 2006. Annual Review of Ecology Evolution and Systematics 37:373-404.
Jones et al. 2012. Global Ecology and Biogeography 21:716-724.
Mieszkowska et al. 2014. Journal of Marine Systems 133:70-76.
Philippart et al. 2011. Journal of Experimental Marine Biology and Ecology 400:52-69.
Poloczanska et al. 2008. Ecology 89:3138-3149.
Southward. 1991. Journal of the Marine Biological Association, U.K. 71:495-513.
Southward and Crisp. 1954. Journal of Animal Ecology 23:163-177.
Southward et al. 1995. Journal of Thermal Biology 20:127-155.
Wethey et al. 2011. Journal of Experimental Marine Biology and Ecology 400:132-144.

3.2. The above is pertinent as in several places the ms talks about changes in the N Atlantic, whereas really this should state NW Atlantic. The authors could point out the differences between the two sides (see Jenkins et al 2008 Ecology). Due to glacial recolonisation processes most of the shore biota in the Gulf of Maine are northern species that probably recolonised from Europe via Iceland and Greenland. There are no southern species to take the place of those declining, in contrast to the NE Atlantic where similar warm water species are advancing or increasing and taking the place of cold water functional equivalents (P.depressa replacing P.vulgata, Chthamalus spp replacing Semibalanus, trochids replacing L.littorea). Thus the food webs of New England/Canada are particularly vulnerable to disruption as there are no warm water substitutes to come and fill a void. With the barnacles, the local Chthamalid is probably unable to get around Cape Cod to fill any gaps and there are no trochids or patellids. This is an exciting extra point to emphasize. (see above listed work for references - mostly in Hawkins et al 2019 Chapter 2 in Interactions in the Marine Benthos, CUP)

Reviewer 1 raised the same point. See our reply to comment 1.10.

3.3. How typical is the study area of the rest of the coast? This needs emphasizing with evidence.

*Sentence added on lines 223-225, which reads, “The species composition of invertebrates and algae at these sites is typical of sheltered rocky shores in the north Atlantic⁴².” Superscript reference is Lewis (1964, *The Ecology of Rocky Shores*).*

3.4. Do you have any data pre-1997 for this coastal area that could even qualitatively stretch back further?

Excellent question and something we hope to explore in another manuscript. We know of sampling done on Swan’s Island during the 1960s and 1970s as part of a summer field course. Sampling was not done at our sites. These data were reported as biomass per m⁻², which was the norm for European marine biologists during the 1960s (the instructor for the course was a Swedish marine biologist). We have access to these data, but in order to compare the earlier data to our data, which are done as counts per 0.25 m⁻², requires reconciling the two different sampling procedures. This requires re-sampling the 1960-1970 sites using both methods. This has been on our “to-do” list for awhile.

3.5. There is a very exciting strong peak around 2006 in most of the focal species - and whilst the overall trend is downward, this does show the capability of the system to bounce back. Do you have any ideas what caused this? It should certainly be discussed somewhere. If the dots were joined up it would be a fluctuating curve and the emphasis on % declines per year suggests a monotonic trend which is not the case. *Semibalanus* in particular typically has very big recruitment years where there is very good match of larval release and the phytoplankton bloom (first spotted by Connell 1961 in *Ecological monographs*; see also Hawkins and Hartnoll, 1982 *JEMBE*). Such peaks in recruitment are not unknown in *Mytilus*. Only one good recruitment in a lifespan (once every 5-6 years??) is sufficient for persistence. The concordance in all these different species suggests some kind of mesoscale or local scale driver - any ideas?

*We did notice the peaks in several species in 2006 when we undertook the first analyses of the data and also suspected some kind of mesoscale driver. However, we cannot point to anything definitive. The suggestion of a link to a good phytoplankton bloom and barnacle recruitment is an excellent idea and worth exploring in more depth. However, this does not explain the peaks in abundance for *N. lapillus*, *L. littorea* and *L. obtusata*.*

While not raised by the reviewer, we would emphasize there is no evidence the peaks are a sampling artifact. The same three people (Petraitis, Dudgeon and Rhile) have been doing all of the sampling starting in 1996. Once we noticed the peak, we immediately checked our sampling notes, which include contemporaneous comments and observations about local conditions and natural history. There is no record that we thought the counts were larger than “normal” as we were collecting the data.

We have added several sentences (lines 109-112) to comment on the peaks, which read, "There are also clear peaks in 2006 in abundances of N. lapillus, L. littorea, and L. obtusata, and in barnacle recruitment (Figs 2d, 2a, 2c, and 2f, respectively). These are more likely due to mesoscale processes rather localized conditions, but we have no evidence for a potential driver."

3.6. What are the consequences of recruitment for adult cover of Mytilus and Semibalanus - is there any density dependent feedback as shown by Jenkins et al 2008 (J.anim. ecol). Could these data be added?

We have those data, but we think adding those data and their analyses would change the focus of the manuscript. We would prefer to keep the present focus on the parallel declines of gastropods, mussels and barnacles.

3.7. Figure 1 is a very dense figure and would benefit from labelling of axes (what units), some labels on graphs (nos, recruitment etc) and referring to different letters for boxes in the legends. Is March the winter minimum temperature - say so in legend?

We have split Figure 1 into two figures (see new Figures 1 and 2), and have made the suggested changes to labels, etc.

3.8. Climate strictly refers to the physical regime of temperature, precipitation, winds etc. Ocean acidification is a co-variate of climate change as both are driven by greenhouse gas Carbon dioxide - but strictly it is a change in biogeochemical processes - you can lump both as global change. This also begs the question is aragonite saturation driving these changes in ecology or just reflecting climate change and co-varying with it (temperature and solubility for instance?). You do say it is general proxy - bit more clarification or qualification would help, perhaps?

Reviewer 1 raises the same point (see our reply to comments 1.1, 1.3 and 1.5).

3.9. L. obtusata lives in a damp refuge - canopy algae - might this explain the lack of trend due to facilitation providing a refuge. In this worth commenting on? (see Moore et al 2007 MEPS for work on Northern limpets under fucoids)?

We are very uncomfortable with this suggestion. We do not present data or analyses that would support a speculation on how canopy cover may buffer L. obtusata from environmental conditions.

However, we would note that we do have the data to test directly and experimentally the hypothesis that canopy algae buffer not only L. obtusata but also other species from environmental effects. Recall that we used only the data from control plots in this manuscript, and we also have similar data from cleared plots, as Reviewer 3 is well aware from seeing our presentations at numerous international meetings. If canopy cover buffers gastropods and other species, there should be a clearing by time interaction. We already are working on this manuscript.

3.10. Fucoids as cold water species may also die as they are susceptible to bleaching events in warm years as adults and also when recruiting. The die off of grazers may enable persistence, as would die-off of *Nucella* as denser barnacles facilitate recruitment of *Fucus*. The interesting thing is the aphasia of these responses to climate. There is some evidence of *Fucus* declines at their southern limits in Europe due to both greater physical stress and more grazing from all those extra clades (patella, trochids) missing in New England (Ferriera et al 2015 MEPS).

*This manuscript is limited to changes in invertebrates and does not include our extensive data and published papers on recruitment and abundance of *Fucus vesiculosus* and *Ascophyllum nodosum*. Also note comment 3.10 is asking a similar question about feedbacks that was raised in comments 3.6 and 3.9. Please see our replies to those comments.*

3.11. L28: - other refs? These refs are mainly about OA - not climate

We added a climate review paper to the sentence on lines 28-30. The reference is Helmuth et al., 2006, which is superscripted reference 5. Reviewer 3 is the senior author on this reference. Also see our reply to comment 3.1.

3.12. L54: - is the decline linear - or a line through a wobble - is % decline per year a valid metric?

We are not sure what the Reviewer is asking here because “line through a wobble” is a subjective statement. Percent decline per year is a valid metric. Recall we used a Poisson model to fit the data and this implies the equation: $Y = \exp(a + bt)$ where $Y = a$ discrete count, $a =$ intercept, $b =$ slope, and $t =$ time in years (see Supplemental Table 1). Percent change per year is defined as: $Y(t+1)/Y(t)$. By substitution, percent change equals $\exp(b)$, which is what is reported in the text and Supplemental Table 2. Finally, we note that the declines are well supported because the credible limits of our Bayesian estimates for b (the slopes) do not include zero.

3.13. L76-78: Any links to stratification of Gulf of Maine and ultimately NAO?? Windier late winters could mean messier and less strong phytoplankton and thus less good match with larval release (there is good evidence for this in *Semibalanus*). Less stratification could also mean weaker fronts retaining larvae??

*We looked at NAO but found no correlations. These analyses were not included in our paper because the original submission was to Nature Ecology & Evolution, which has a very strict limit of 1500 words. We would like to point out Hawkins (Reviewer 3) was the senior author on Mieszkowska et al. (2014), which also found no correlation of *S. balanoides* abundance with NAO.*

3.14. L94: - any correlation (with lags?) of *Nucella* with barnacles and mussels??

*Supplemental Table 5 in the original submission shows weak correlations of *N. lapillus* abundance with recruitment of barnacles and mussels. New Supplemental Table 8 with correlations based on residuals shows no significant correlations. Also see our reply to comment 1.8.*

3.15. L110: swathes

Changed to “swathes” but please note the preferred plural in the US is “swaths” as originally written.

3.16. L124: - although the grazing regime might get back to pre-1860, the system as a whole will not due to all the other climate driven changes. This is an important point as this is a non-native driven system from the top include Green(shore) crabs. Perhaps just rephrase little.

*We have re-written this section. Lines 179-184 read, “Mussels have already disappeared from many intertidal shores throughout the North Atlantic^{20,28}; a loss that has been widely attributed to the recent rise in the predatory green crab, *Carcinus maenas*^{29,30}. Yet, our results suggest recruitment failure associated with processes in the water column is also playing an important role in the decline of mussels throughout the North Atlantic.”*

We know of anecdotal evidence that mussels are declining throughout the north Atlantic. Colleagues in Ireland, Scotland, and Germany, as well as several colleagues who work in the Gulf of Maine have told us that well-established mussel beds are disappearing. Green crabs are not undergoing expansion in all of those locations.

Superscripted references 20, 28, 29 and 30 are Sorte et al. (2017), Commito et al. (2019), Beal et al. (2018), and Roman (2006), respectively.

3.17. L306: typo - author missing in reference???

Fixed.

REVIEWERS' COMMENTS:

Reviewer #1 (Remarks to the Author):

This is a second round of review and I was one of the reviewers of the first round. I enjoyed reading the rebuttal and new manuscript, and agree that the authors have made substantial changes, which have improved the manuscript. They have addressed all the issues raised by the original review and I do not have any further comments. I recommend this manuscript for publication.

(Just one return discussion note on point 1.3 of the rebuttal - the reviewer was making the suggestion of using salinity in the assumption that there was more salinity data available, and so the comment that the reviewer is being disingenuous does not hold given that they did not know the amount of salinity data that might or might not be available. It was made simply as a suggestion in the hope that, given that it is often an easier parameter to measure, there would be more data, much like there is with temperature).

Reviewer #3 (Remarks to the Author):

The authors have done an excellent and thoughtful job answering my and the other referee's queries and comments. I agree with them that the short format of the journal means all nuances cannot be covered.

I had no substantive criticisms of the first version and have none for this version; thus I would like to see this paper published now. They have addressed the concerns and improved the ms.

I apologize for using the medium of refereeing in having a bit of a discussion with the authors. I will point out that we did extend the Southward times series with counts made myself from 1997 to the mid 2000s (in Phillipart et al 2011) and beyond at the one site studied longest by Southward (in Mieszkowska et al 2014) - and the interesting thing was the persistence of *Semibalanus* (even though abundance was lower) - just the odd good recruitment every 3-5 years seems enough to keep range edge populations topped up. But this is not relevant to this paper.

There is just one minor caveat I would like to see inserted and my original comments were perhaps a little flippant ("line through a wobble"). Perhaps around line 70-75 a few words or sentence could be added to comment on the nature of the decline (fluctuations around a downward trend).

I accept there is an overall downward trend, but this is not uniformly monotonic: there are fluctuations around the downward trend, with some concordance in some species, but not others, as discussed later in the MS. My point is that declines per year, whilst having some predictive power and give the big picture, can obscure a more complex and messier picture: species can and do bounce back (even if this is temporary). The extent of fluctuations in recruitment of *Mytilus* and *Semibalanus* are dampened in Figure 1 due to the entirely appropriate transformations used - recruitment variation tends to operate on non-arithmetic scales.

Minor point: I would like to see units given on the axes in Fig 2 (e.g. temperature).

This is a nice piece of research and the authors are to be commended on their persistence in keeping these studies going for so long.

REVIEWERS' COMMENTS:

Reviewer #1 (Remarks to the Author):

This is a second round of review and I was one of the reviewers of the first round. I enjoyed reading the rebuttal and new manuscript, and agree that the authors have made substantial changes, which have improved the manuscript. They have addressed all the issues raised by the original review and I do not have any further comments. I recommend this manuscript for publication.

(Just one return discussion note on point 1.3 of the rebuttal - the reviewer was making the suggestion of using salinity in the assumption that there was more salinity data available, and so the comment that the reviewer is being disingenuous does not hold given that they did not know the amount of salinity data that might or might not be available. It was made simply as a suggestion in the hope that, given that it is often an easier parameter to measure, there would be more data, much like there is with temperature).

Response: We apologize for our comment about the reviewer being disingenuous. Salinity data are available from offshore buoys but we are unaware of any long-term data at onshore locations near our study sites. As we discussed in our original responses, data of offshore salinity are not likely to provide good proxies for near-shore salinities. We agree data for near-shore salinities near our sites, if available, would very useful in expanding our analyses.

Reviewer #3 (Remarks to the Author):

The authors have done an excellent and thoughtful job answering my and the other referee's queries and comments. I agree with them that the short format of the journal means all nuances cannot be covered.

I had no substantive criticisms of the first version and have none for this version; thus I would like to see this paper published now. They have addressed the concerns and improved the ms.

I apologize for using the medium of refereeing in having a bit of a discussion with the authors. I will point out that we did extend the Southward times series with counts made myself from 1997 to the mid 2000s (in Phillipart et al 2011) and beyond at the one site studied longest by Southward (in Mieszkowska et al 2014) - and the interesting thing was the persistence of Semibalanus (even though abundance was lower) - just the odd good recruitment every 3-5 years seems enough to keep range edge populations topped up. But this is not relevant to this paper.

There is just one minor caveat I would like to see inserted and my original comments were perhaps a little flippant ("line through a wobble"). Perhaps around line 70-75 a few words or sentence could be added to comment on the nature of the decline (fluctuations around a downward trend).

I accept there is an overall downward trend, but this is not uniformly monotonic: there are fluctuations around the downward trend, with some concordance in some species, but not others, as discussed later in the MS. My point is that declines per year, whilst having some predictive power and give the big picture, can obscure a more complex and messier picture: species can and do bounce back (even if this is temporary). The extent of fluctuations in recruitment of *Mytilus* and *Semibalanus* are dampened in Figure 1 due to the entirely appropriate transformations used - recruitment variation tends to operate on non-arithmetic scales.

Response: We agree that there is “wobble” and the observed mean yearly averages for abundances and recruitment do not fall in tight patterns around the estimated rates of decline. In the original and the revised versions of the manuscript, we discussed several possible causes for the variation in the last paragraph of the Results section. We see no need to add an additional comment around lines 70-75 (the end of the first paragraph of the Results section).

We agree with the reviewer that recruitment of mussels and barnacles is highly variable, but we do not think it “operate[s] on non-arithmetic scales.” We think a better choice of words would be variation in recruitment is best captured by using a log-scale. We would also point out Poisson regression, which we used, assumes the expected value of the dependent variable is modeled with the independent variables on a log-scale. The plots for mussel recruitment were done using a log-scale improve the clarity of the presentation. The model for barnacle recruitment would not converge using untransformed data, and so the plots for barnacles were done using a transformed scale to preserve the model predictions in the appropriate scale.

Minor point: I would like to see units given on the axes in Fig 2 (e.g. temperature).

Response: Units added as requested.

This is a nice piece of research and the authors are to be commended on their persistence in keeping these studies going for so long.

Response: Thank you!